# High-Luminescence Electrospun Polymeric Microfibers In Situ Embedded with CdSe Quantum Dots with Excellent Environmental Stability for Heat and Humidity Wearable Sensors

**DOI:** 10.3390/nano12132288

**Published:** 2022-07-03

**Authors:** Chenyu Zhu, Qiao Wang, Guorong Sun, Suo Zhao, Yao Wang, Tonghui Li, Xianglong Hao, Mikhail Artemyev, Jianguo Tang

**Affiliations:** 1Institute of Hybrid Materials, National Centre of International Joint Research for Hybrid Materials Technology, National Base of International Science & Technology Cooperation on Hybrid Materials, Qingdao University, 308 Ningxia Road, Qingdao 266071, China; zhuchenyuzcy@163.com (C.Z.); wwwqqq721@163.com (Q.W.); sgr0505@126.com (G.S.); zhaosuo2022@163.com (S.Z.); wangyaoqdu@126.com (Y.W.); litonghuilth@163.com (T.L.); hxl316439623@163.com (X.H.); 2Research Institute for Physical Chemical Problems of the Belarusian State University, 220006 Minsk, Belarus

**Keywords:** CdSe, polymer, hybrid fibers, photoluminescence thermal sensor

## Abstract

In this paper, hydrophobic luminescent CdSe quantum dots are successfully dispersed in a mixture of styrene and methyl methacrylate through the oleic to methacrylic acid ligand exchange. Further in situ solution polymerization of the quantum dots in a mixture of styrene and methyl methacrylate followed by electrospinning allowed us to prepare luminescence hybrid styrene-co-methyl methacrylate fibers embedded with quantum dots. CdSe@P(S+MMA) hybrid fibers with 27% quantum yield showed excellent moisture, heat and salt resistance with a photoluminescence output below 120 °C. When dry heated, the hybrid fibers of the fluorescence signals decreased with temperature to 79%, 40%, 28%, 20% and 13% at 120 °C, 140 °C, 160 °C, 180 °C and 200 °C, respectively, due the to the chemical degradation of CdSe QDs. Such hybrid fibers show the potential to manufacture wearable moisture- and heat-sensing protective clothing in a 120–200 °C range due to the thermal-induced quenching of quantum dot photoluminescence.

## 1. Introduction

Quantum dots (QDs) are regarded as outstanding light-emitting materials which have excellent photophysical properties, including high-fluorescence quantum yield and a unique size-controlled emission spectral range [1,2,3]. Moreover, QDs have broad absorption and narrow emission, with the advantages of wide color gamut, enhanced chromaticity and high intensity [4,5]. In the last several decades, QDs have been utilized in biosensors [6,7], digital communication [8], detectors [9], photoelectricity, catalysis, light-emitting diodes, solar cells, and photodetectors [10,11,12,13,14,15,16]. However, core-type QDs contain a large amount of surface defects resulting in photothermal instability, photoluminescence (PL) quenching, and bad chemical stability.

Inorganic semiconductor materials with similar crystalline lattices but larger band gaps to wrap on the surface of QDs [17], or monolayers of organic ligands (carboxylic acids, organic phosphorus, or organic amines) [18] have been widely used to reduce the surface defects of QDs. Embedding QDs into thin organic and inorganic films is an important step towards their various practical applications. Spin coating, layer-by-layer assembly and solution-mixing methods have been previously developed to prepare QDs doped with luminescent thin polymer films [19,20,21]. However, the chemical or physical incompatibility between polymer matrixes and QD surface ligands makes it difficult to form a homogeneous film and affects emission characteristics. Previously, Hu et al. used the ligand exchange method to improve the electron transfer efficiency between CdSe QDs and electron transport layer [22].

Recently, electrospinning has garnered attention in sensing for glucose, H_2_O_2_, dopamine, ascorbic acid, uric acid, neurotransmitters, biomedical applications (wound dressing and healing; skin, nerve and bone tissue engineering; and drug delivery systems), water treatment, energy harvesting, and storage applications [23]. Moreover, electrospinning has also been found to be a versatile and effective technique to fabricate fibrous membranes with large surface areas and three-dimensional porous structures for rapid and highly sensitive sensors [24,25,26,27]. So far, embedding QDs into polymer nanofibers through electrospinning not only brings unique optical and electrical properties to the polymer fibers, but also improves the fluorescence stability of QDs [28,29,30,31,32,33,34,35]. There was no significant influence of QD content onto the diameter of the hybrid fibers [36]. Recently, growing efforts have been made to integrate QDs into electrospun fibers for novel optical sensors with good reproducibility, excellent photochemical stability, and high sensitivity. Nowadays, by introducing sensing probes or molecular recognition sites, electrospun nanofibers are utilized as various fluorescence sensors for metal ions, gas and pH [37,38,39,40,41]. However, to the best of our knowledge, there are no reports on fabricating wearable moisture- and heat-sensing protective clothing based on electrospun fibers embedded with CdSe QDs.

Herein, we use high-quality CdSe QDs prepared by an optimized hot injection method as the starting luminescence material. Then, oleic acid surface ligands are exchanged with methacrylic acid to disperse CdSe QDs easily and uniformly in the polymer matrix. The in situ embedding of CdSe QDs is achieved by the solution polymerization of a styrene/methyl methacrylate mixture followed by forming luminescent QD-based hybrid fibers with the electrospinning technique. The final hybrid fibers show excellent optical properties, moisture resistance, and heat and salt resistance below 120 °C. The intensity of the fluorescence signal from the CdSe@P(S+MMA) hybrid fibers decreases with the gradual deterioration in the external humid and hot environment, allowing the direct detection of moisture and heat. Such hybrid fibers may have great potential to manufacture wearable moisture- and heat-sensing protective clothing.

## 2. Materials and Methods

### 2.1. Chemicals

Cadmium oxide (CdO, 99.99%), selenium powder (Se, 99.99%), 1-octadecene (1-ODE, 90%), trioctylphosphine (TOP), oleylamine (OAm, 80–90%), oleic acid (OA, 90%), styrene (St), methyl methacrylate (MMA), and methacrylic acid (MAA) were purchased from Aladdin, Shanghai, China. Trichloroethylene (C_2_HCl_3_), isopropanol, methanol, and azodiisobutyronitrile (AIBN) were purchased from Sinopharm Chemical Reagent Co., Ltd., Shanghai, China. Except styrene, which was distilled under reduced pressure, all chemicals were used as received and without further purification.

### 2.2. Characterization

The UV-vis absorption of the QD solution and the CdSe@P(S+MMA) hybrid fibers was recorded using a PerkinElmer Lambda 750S spectrophotometer in the 200–2300 nm spectral range using a glass cuvette with a 1 × 1 mm^2^ optical path (Craic 20/30, PerkinElmer, Waltham, MA, USA). Photoluminescence (PL) spectra and fluorescent optical microscopic photographs were measured with a Cary Eclipse fluorescent spectrometer (FLS1000, Edinburgh Instruments Ltd., Edinburgh, UK) and an OLYMPUS BX41 universal microscope with a UV lamp emitting in the 380–430 nm spectral range (CRAIC, San Dimas, CA, USA), respectively. Fourier transform infrared (FTIR) spectra were recorded on a Nicolet 5700 infrared spectrophotometer in the 4000–400 cm^−1^ spectral range (Nicolet Instrument Corporation, Fitchburg, MA, USA). The morphology of the hybrid fibers was characterized by scanning electron microscopy (SEM, JEOL 6460) (JEOJ, Kyoto, Japan). STEM-EDS elemental mapping was completed on a Tecnai G2 F20 S-TWIN instrument (JEOL, Kyoto, Japan). Transmission electron microscopy (TEM) was recorded with a JEM-2100HR (JEOL, Kyoto, Japan) instrument operating at 200 kV. Electrospinning was performed on a DFS-01 instrument in the 0–30 kV range (Beijing kaiweixin Technology Co., Ltd., Beijing, China). The PL quantum yield was recorded on a Cary Eclipse fluorescent spectrometer (FLS1000, Edinburgh Instruments Ltd., Edinburgh, UK). XPS spectra were obtained by a XSAM-800 Kratos spectrometer (JEOJ, Ltd., Japan).

### 2.3. Preparation of CdSe@P(S+MMA) Hybrid Fibers

CdO (2.4 mmol) was dissolved in a mixed solution of 2.5 mL 1-ODE and 2.5 mL OA in a three-neck flask. The reaction mixture was heated up to 100 °C under vacuum for 20 min, and then heated to 250 °C in the nitrogen atmosphere for 5 min to obtain the Cd oleate as the precursor. Se powder (1.2 mmol) was dissolved into 1 mL TOP under ultrasonic treatment at room temperature in the inert atmosphere. Then, 0.4 mL TOP-Se solution, 0.6 mL TOP, and 9 mL OAm were introduced into a three-neck flask and degassed under vacuum at 100 °C for 20 min with stirring. When the temperature reached 275 °C, 1 mL of cadmium oleate solution was rapidly injected into the reaction mixture under an N2 atmosphere, stirred for 4 min, and then cooled down to room temperature. The obtained CdSe QDs were precipitated with methanol, centrifuged at 8000 rpm for 5 min, and dried.

Then, 6.65 mmol of dry CdSe QD was dissolved in 20 mL of a mixture of styrene and methyl methacrylate (volume ratio of St:MMA = 10:0, 9:1, 8:2, 7:3, 6:4); 70 μL of methacrylic acid was added and kept for 12 h to complete the oleic to methacrylic acid ligand exchange. Then, 0.055 mmol of AIBN initiator and 10 mL of methyl acetate as a solvent were added and heated on the oil bath at 100 °C for 2 h for pre-polymerization, followed by 12 h at 130 °C to complete polymerization, before being cooled to room temperature to obtain the solution of polystyrene-co-methyl methacrylate with the in situ embedded CdSe QDs.

The obtained CdSe@P(S+MMA) polymer solution was loaded into 5 mL plastic syringes with a stainless-steel needle. During the electrospinning, the aluminum foil served as the positive electrode with the applied voltage of 13 kV. The jet velocity was kept at 0.05 mL/min and the distance between the spinneret tip and the collector was 15 cm. The experimental design of the luminescent hybrid fibers is shown schematically in Figure 1.

## 3. Results

### 3.1. Analysis of Copolymer Properties with Different Volume Ratios

A method for preparing the polystyrene–methacrylate copolymer P(S+MMA) in situ for CdSe quantum dots has been proposed. We chose mixed solutions of styrene and methyl methacrylate of different volume ratios. In order to demonstrate the fluorescence properties of the obtained P(S+MMA) packaging with the CdSe quantum dots, the hybrid fibers with different styrene and methyl acrylic ratios were characterized by fluorescence spectroscopy and UV absorption spectroscopy. As shown in Figure 2a,b, when the volume ratio was 7:3 the fluorescence intensity of the luminescence system was the highest, at more than twice the luminescence peak of the other proportional mixed copolymers, and masked the original luminous peak of styrene and methyl methacrylate copolymers. Thus, in future experimental processes, we will choose a styrene: methyl methacrylate volume mixing ratio of 7:3. Figure 2c,d demonstrate the absorption and PL spectra of the CdSe@P(S+MMA) hybrid fibers with different CdSe QD contents. The data of Figure 2 show that the magnitude of PL and the optical absorption signals were proportionally increased with the QD content.

SEM images of the hybrid fibers with different QD contents are shown in Figure 3a–f. The morphology of the CdSe@P(S+MMA) hybrid fibers was similar to that of the original P(S+MMA) ones, which indicates that the QD content had no marked influence on the hybrid fiber morphology. However, the 5% and higher QD content affected the polymerization process, resulting in consequent problems with spinning procedures. Figure 3g–l demonstrate the fluorescence microscopic images of the CdSe@P(S+MMA) hybrid fiber networks on the glass substrate under UV excitation. The fiber networks exhibited bright red emission, which indicates that the CdSe QDs were introduced homogeneously compared to the whole hybrid fibers. Figure 3h shows that 1% weight content was too low to obtain strong luminescence from the hybrid sample. However, the sample with 5% QD content demonstrated not-uniform fiber morphology. Considering that for a damp heat-sensing application a high-fluorescence signal from the QDs is required, 4% wt QD content was selected for further manipulations.

### 3.2. Luminescence and Composition of Hybrid Fibers

Next, we compared the optical properties of the hybrid fibers and the corresponding QD colloidal solutions with and without the polymers. Figure 4a,b show the absorption and emission characteristics of CdSe QDs in trichloroethylene at room temperature. The first excitonic absorption peak was around λ ≈ 628 nm with a Stokes shift of 8 nm. The emission peak of the CdSe@P(S+MMA) solution was at λ ≈ 633 nm, while for hybrid fibers this was around λ ≈ 646 nm. It should be noted that the emission peak in the hybrid fibers was slightly redshifted as compared to the CdSe@P(S+MMA) solution due to both the change in the local refractive index and the aggregation of the nanocrystals. Panel (c) compares the relative PL photostability of QDs in solution, CdSe@P(S+MMA) solution, and hybrid fibers under prolonged UV irradiation. Within 2 h, the intensity of the PL signal for the colloidal QDs dropped to 63%, the CdSe@P(S+MMA) solution to 91%, and the hybrid fibers to 96% of the initial level. The PL quantum yield remained around 27–28% for all three samples. Figure 4d shows the corresponding PL decay times for the QD colloids and hybrid fibers. The average decay time increased from 6 to 20 ns when going from colloidal QDs to hybrid fibers. All these data demonstrate that the polymeric matrix of hybrid fibers effectively improves the photostability of QDs without affecting their quantum yield.

Figure 5a–d show the STEM-EDS elemental mapping of the hybrid fibers. The Cd and Se atoms are distributed homogeneously over the whole hybrid fiber. The EDS elemental analysis of the hybrid fiber surface gives the atomic content of Cd and Se as 0.08% and 0.14%, respectively. As shown in Figure 5e, the TEM image of the hybrid fiber shows that the fiber surface was smooth, and the local enlarged image (f) demonstrates that the QDs were evenly distributed on the fiber surface. However, we believe that most of the QDs were located inside the hybrid fiber. In order to verify the spatial distribution of the QDs inside the hybrid fiber, we performed a STEM-EDS test of the fiber tip. The corresponding STEM-EDS elemental maps in Figure 5g–l show the distribution of the C (c), O (d), Se (e) and Cd (f) atoms. The data of Figure 5g–l confirm that most CdSe QDs were located in the middle of the co-polymer hybrid fiber.

In order to verify the chemical composition of the polymer matrix we carried out FTIR tests on pure P(S+MMA) and CdSe@P(S+MMA) fibers. Figure 6a shows that there were a number of characteristic bands of P(S+MMA). The bands at 697 cm^−1^ and 757 cm^−1^ are the bending in-plane vibrations of the monosubstituted C-H group of the benzene ring, and the bands at 1120 cm^−1^ and 1180 cm^−1^ relate to the stretching vibrations of the ester C-O-C group. Moreover, the band at 1450 cm^−1^ is the bending in-plane vibration of the O-H group from the carboxylic acid. The peak at 1720 cm^−1^ is related to the tensile vibration of the ester C=O group, and the band at 2930 cm^−1^ is the tensile vibration of the O-H group from the carboxylic acid.

A typical XPS wide scan spectrum of CdSe@P(S+MMA) hybrid fiber is shown in Figure 6b. The XPS spectrum also shows that the sample was mainly composed of Cd, Se, C and O elements. Figure 6c,d show the narrow scan spectra for the Cd3d and Se3d regions, respectively. The two strong peaks at the Cd region at 411.4 and 404.5 eV correspond to the Cd3d3/2 and Cd3d5/2 binding energies. The peak at 54.3 eV measured in the Se energy region is attributed to the Se3d transition.

### 3.3. Sensing Properties of Hybrid Fibers

As shown in Figure 1c, the hybrid fiber was placed under different conditions to test its dielectric resistance. Figure 7a,b show that dropping aqueous solutions of different metal salts onto CdSe@P(S+MMA) hybrid fiber film for 30 min does not sufficiently change their fluorescence output. The fluorescence intensity of the hybrid fiber soaked in the metal salt at room temperature remained above 95%. Soaking the hybrid fibers in the aqueous solutions of different metal salts at 60 °C (Figure 7c,d) resulted in partial PL quenching accompanied by a decrease in optical density at the first exciton band. We assign this effect to the chemical degradation or washing out of QDs located close to the fiber surface. However, it is worth mentioning that the fluorescence intensity of the hybrid fiber remained above 80%. Dry heating the hybrid fibers at different temperatures below 80 °C for 30 min (Figure 7e,f) did not lead to PL quenching. At 120 °C, the PL quenched only 20%, which demonstrates the good thermal stability of the CdSe QDs embedded into the hybrid fibers.

In order to further explore the possible application of hybrid fibers in damp heat sensing, as shown in Figure 1c, the hybrid fiber was placed under different conditions to test its dielectric resistance. We carried out the fluorescence tests under dry and damp heating conditions (the result are shown in Figure 8a–d). When the sample was fumigated with water vapor at 60 °C, the fluorescence quenched only 2%. With the increase in temperature, the fluorescence signal from the hybrid fibers decreased to 82%, 74% and 64% at 70 °C, 80 °C and 90 °C, respectively. When the steam temperature rose to 100 °C, the fluorescence signal decreased to 55% of the initial level, accompanied with a decrease in optical density. Under dry heating, the fluorescence signal further decreased to 79%, 40%, 28%, 20% and 13% with increasing temperatures of 120 °C, 140 °C, 160 °C, 180 °C and 200 °C, respectively, and was almost quenched at over 200 °C due to the chemical degradation of the CdSe QDs. Therefore, such hybrid fibers show the potential to manufacture wearable moisture- and heat-sensing protective clothing in a 120–200 °C range.

## 4. Conclusions

In summary, hybrid luminescent CdSe@P(S+MMA) fibers were successfully fabricated via electrospinning technology using polystyrene-co-methyl methacrylate in situ embedded with hydrophobic CdSe QDs. The as-prepared hybrid fibers showed bright CdSe QD emission with the QY of the initial hydrophobic QDs. When the QD content exceeded 4% wt the polymerization process worsened, resulting in a low polymerization rate and problems with electrospinning. The fluorescence signal from the hybrid fibers with 4% wt QD content decreased in humid and hot environments. The hybrid fibers showed good moisture, heat and aqueous salt resistance below 120 °C, demonstrating the potential to manufacture wearable moisture- and heat-sensing protective clothing in the 120 to 200 °C range due to thermal- and moisture-induced PL quenching.

## Figures and Tables

**Figure 1 nanomaterials-12-02288-f001:**
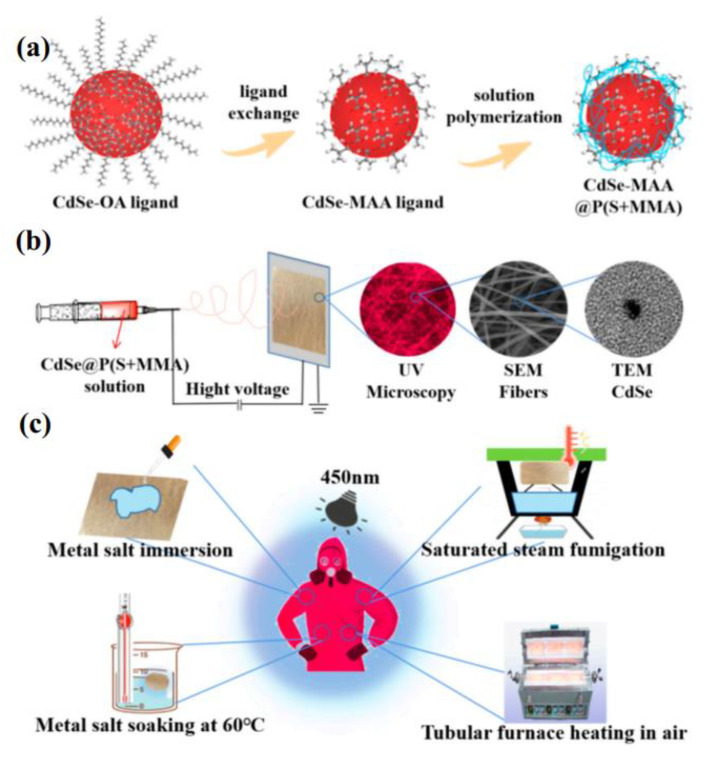
(**a**) Scheme of the oleic acid ligand exchange with methacrylate acid followed by solution polymerization; (**b**) schematic diagram of the electrospinning process; (**c**) possible practical applications of the luminescent electrospun hybrid fibers.

**Figure 2 nanomaterials-12-02288-f002:**
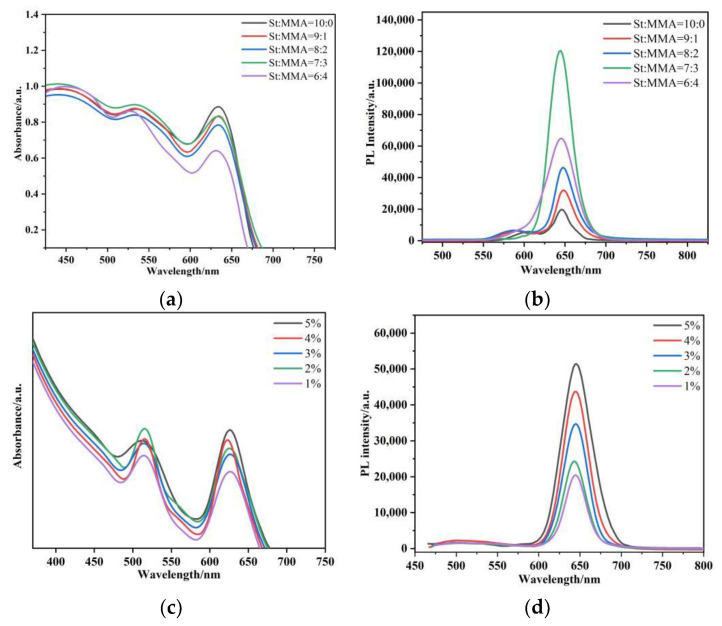
(**a**) Absorption and (**b**) photoluminescence spectra of P(S+MMA) with different volume proportions of packaged CdSe quantum dots. (**c**) Absorption and (**d**) photoluminescence spectra of the CdSe@P(S+MMA) nanofiber network with different CdSe weight content. λ_exc_ = 450 nm.

**Figure 3 nanomaterials-12-02288-f003:**
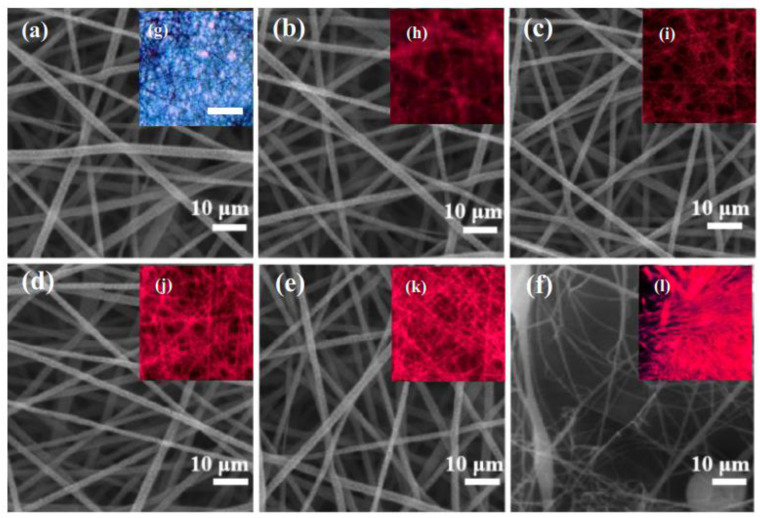
SEM (**a**–**f**) and the microfluorescence images (**g**–**l**) (under UV excitation); CdSe QD content of 0% wt (**a**,**g**), 1% wt (**b**,**h**), 2% wt (**c**,**i**), 3% wt (**d**,**j**), 4% wt (**e**,**k**) and 5% wt (**f**,**l**). The scale bar for (**g**–**l**) is 100 μm.

**Figure 4 nanomaterials-12-02288-f004:**
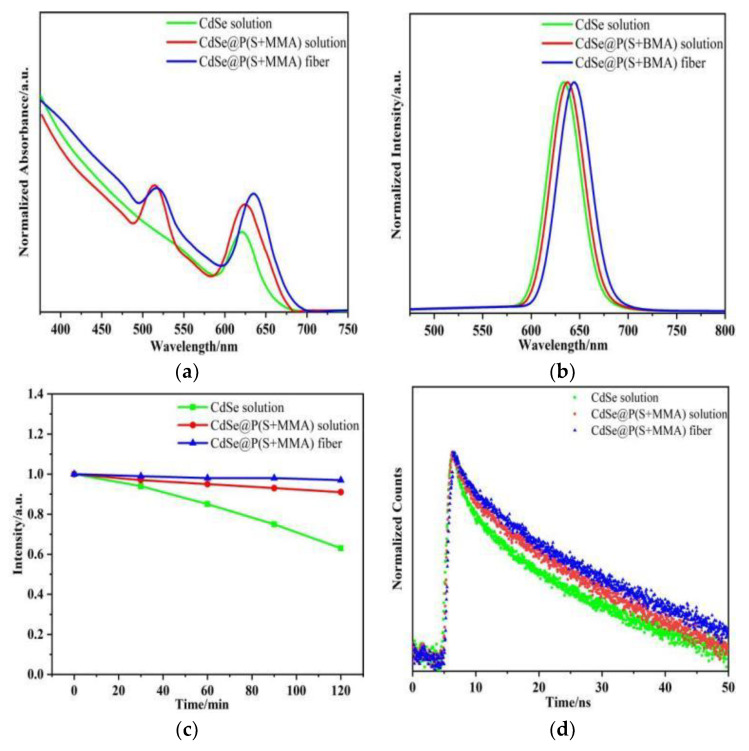
Absorption (**a**) and PL (**b**) spectra of CdSe QDs and CdSe@P(S+MMA) solution in trichloroethylene and CdSe@P(S+MMA) hybrid fibers. (**c**) Photostability of PL signal from CdSe QDs, CdSe@P(S+MMA) solution in trichloroethylene, and CdSe@P(S+MMA) hybrid fibers under 2 h UV lamp irradiation. (**d**) PL decay curves for CdSe QDs and CdSe@P(S+MMA) solution for trichloroethylene and CdSe@P(S+MMA) hybrid fibers. λ_exc_= 450 nm.

**Figure 5 nanomaterials-12-02288-f005:**
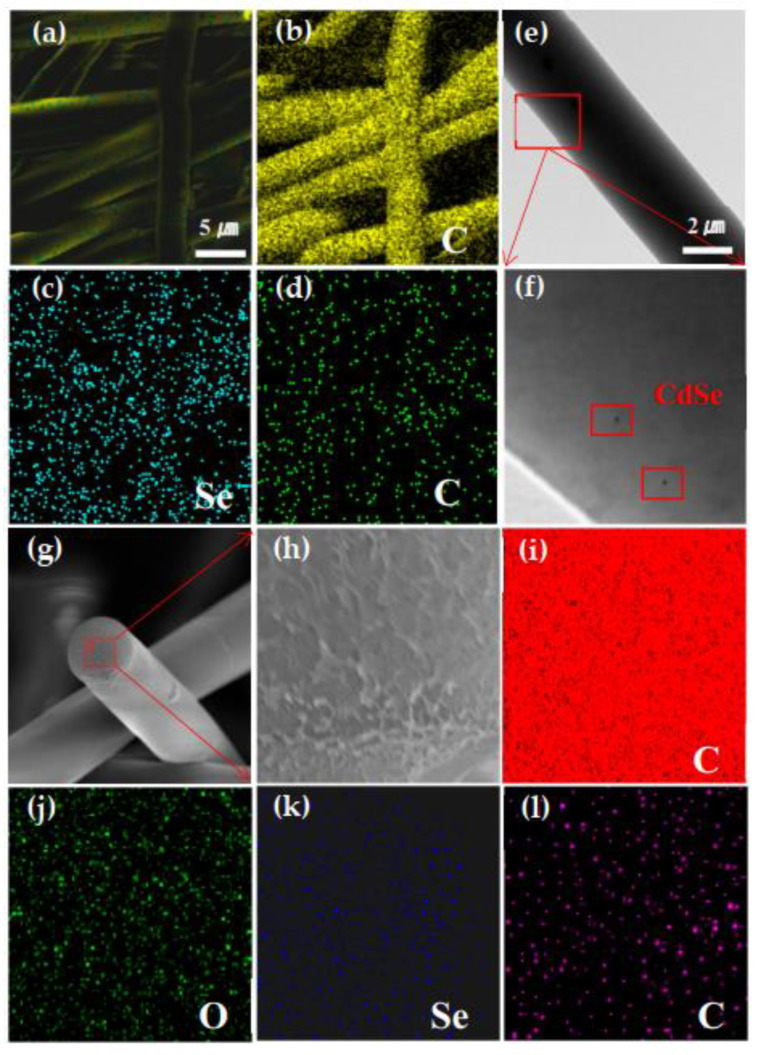
(**a**) EDS layered image of CdSe@P(S+MMA) hybrid fibers. STEM-EDS elemental maps showing the distribution of the C (**b**), Se (**c**), and Cd (**d**) atoms. (**e**) TEM of a single CdSe@P(S+MMA) hybrid fiber. (**f**) Partial enlarged view of selected area on panel (**e**). (**g**) SEM image of CdSe@P(S+MMA) hybrid fiber. (**h**) Partial enlarged view of the fiber tip. STEM-EDS elemental maps showing the distribution of C (**i**), O (**j**), Se (**k**), and Cd (**l**) atoms.

**Figure 6 nanomaterials-12-02288-f006:**
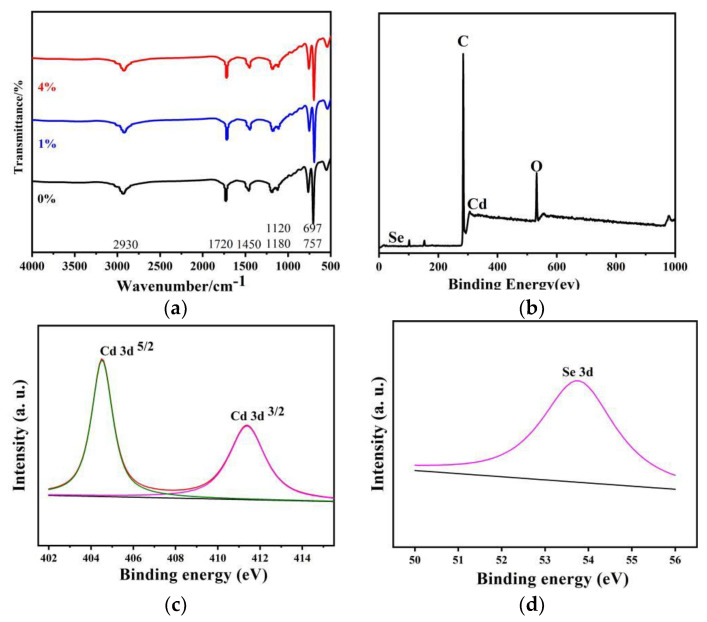
(**a**) FTIR spectra of the P(S+MMA) and CdSe@P(S+MMA) hybrid fibers. (**b**) XPS wide scan spectrum of CdSe@P(S+MMA) hybrid fibers. XPS narrow scan spectrum for Cd 3d (**c**) and Se 3d (**d**) regions.

**Figure 7 nanomaterials-12-02288-f007:**
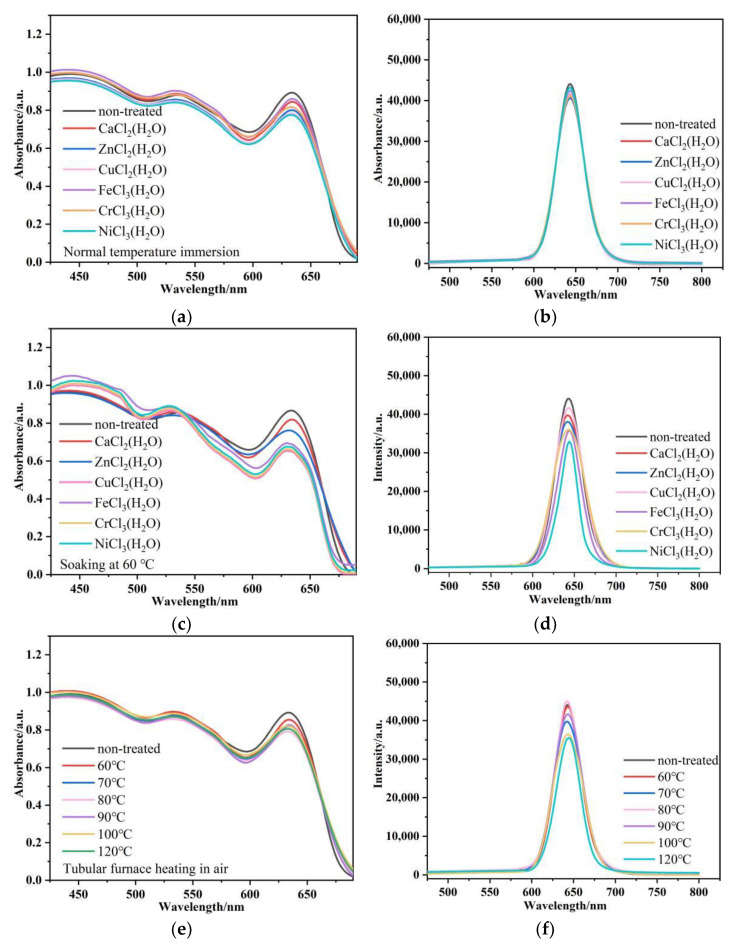
Absorption (**a**,**c**,**e**) and PL (**b**,**d**,**f**) spectra of CdSe@P(S+MMA) hybrid fibers under different conditions: (**a**,**b**) 30 min after the dropping of room temperature; (**g**,**h**) after 30 min with aqueous solutions of different metal salts; (**e**,**f**) after 30 min of dry heating at different temperatures. λ_exc_ = 450 nm.

**Figure 8 nanomaterials-12-02288-f008:**
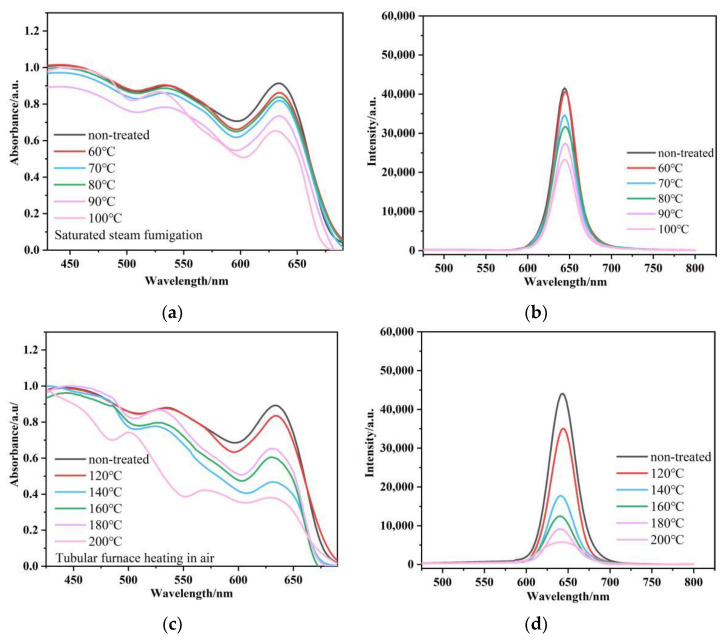
Absorption and photoluminescence spectra of CdSe@Poly(S-MMA) hybrid fibers under different conditions for 30 min; (**a**,**b**) steaming fumigation at different temperatures; (**c**,**d**) dry heating at different temperatures. λ_exc_ = 450 nm.

## Data Availability

All data, models and codes generated or used during the study appear in the submitted article.

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
