# Peer review of "High-Luminescence Electrospun Polymeric Microfibers In Situ Embedded with CdSe Quantum Dots with Excellent Environmental Stability for Heat and Humidity Wearable Sensors"

_nanomaterials, 2022, doi:10.3390/nano12132288_

Round 1

Reviewer 1 Report

This manuscript reports the preparation of P(S+MMA) fibers doped with CdSe quantum dots (QDs), and their photoluminescence and sensing properties towards different conditions were also investigated. The corresponding characterizations of fibers were also conducted by using FTIR, UV, PL, and XPS. The results of this article present significant scientific contributions with respect to the development of chemosensory materials. I feel that major modifications are necessary before publication can be considered.

 (1)   The biggest concern of the reviewer is the synthetic routes of synthesis part. The authors used the P(S+MMA) polymer with different ratios to compare the CdSe@P(S+MMA) hybrid fibers with UV and PL spectra. However, the authors did not mention the synthesis part of P(S+MMA) polymer in section 2.3.

(2)   The second concern of reviewers is the characterization of CdSe QDs in the part of results and discussion. Before discussing the data of CdSe@P(S+MMA) hybrid fibers, the authors need to provide the size, absorption and photoluminescence data, or cite the corresponding reference.

(3)   In abstract, the description is so simple and short, and the main explanations should be added and strengthened.

(4)   In section 2.2, the characterization of XPS and electrospinning (injection conditions) should be given. Besides, how to obtain the fluorescence quantum efficiency? It must also be supplemented.

(5)   In section 2.3, the used amount of reactants should be in mole, not in grams, due to the different molecular weights.

(6)   In lines 100-101, why did the authors need to add 0.6 ml TOF and 9 ml OAm into the system after forming the TOF-Se precursor?

(7)   In line 107, it is not reasonable for the authors to use 70 µl of methacrylic acid instead of OA.

(8)   In the second paragraph of section 2.3, why did the authors not remove the unreacted monomers after polymerization?

(9)   In the whole article, the description of Figure 1(c) was missing.

(10)   In the section 3.1, the statements of Fig. 3(a-f) should be mentioned before Fig. 3(g-l).

(11)   In Figure 2, the title should be (a) absorption and (b) photoluminescence spectra of P(S+MMA) different volume proportions packaged CdSe quantum dots, not (S+MMA). Besides, how to get the 1-5% of Fig. 2(c-d)? In Fig. 2c and 4a, what did the absorption about 520 nm mean?

(12)   In Figures 2 and 4, how to obtain the absorption and photoluminescence spectra of the hybrid fibers?

(13)   In line 140, it should be Fig. 3h, not Fig. 2g.

(14)   In the FTIR analysis (line 204), the fibers possess P(S+MMA) in the system. How to get the band at 2930 cm-1 from the tensile vibration of O-H group from carboxylic acid?

(15)   In the section 3.2, the authors used many different environments to discuss the PL quenching behaviors of CdSe@P(S+MMA) hybrid fibers. However, the statements throughout section 3.2 are so rough that some of the quenching phenomena are not clearly understood. Also, air flow rates or other environmental conditions are not specified. Therefore, the reviewer suggested that this paragraph must be rewritten. Furthermore, the degree of PL quenching must also be quantified.

Reviewer 2 Report

This manuscript reports a new kind of polymer containing CdSe quantum dots. The resulting composite is synthetized in a fiber morphology with potential temperature and humidity sensing applications. The work comprises the synthesis of the materials, the structural characterization, the optical measurements and the test of the sensing performances via the dependence of the absorption and photoluminescence (PL) under different environment conditions (environment salts and temperature). The manuscript is well-written and presented, figures are clear, methods are thoroughly described and conclusions appropriate. Therefore, I can recommend its publication in nanomaterials after addressing the following minor issues:

1. Line 172. Authors claim that the redshift is caused by the change of the local refractive index. Did authors discard the aggregation of nanocrystals as the reason of the redshift?

2. Decay time in Figure 4d is clearly biexponential. I think it could be interesting to comment the origin of the two decay times and also deduce its values.

3. Insets of Figure 7 cannot be seen properly. I recommend the authors to include these insets as separated figures.

4. I think the caption “dielectric properties of hybrid fibers” because no refractive index information is presented. May be “Sensing properties of hybrid fibers” would be more appropriate.

5. In line 226-227 the authors indicate that “We assign this effect to the chemical degradation or washing out QDs located close to the fiber surface”. Is this a limitation of the performance of the composite as sensor or it can be sequentially used with different metal salts?

6. Are the heating and cooling processes carried out in Figure 7 reversible?

7. Is there a maximum temperature where the device can work properly (before damaging)?

Round 2

Reviewer 1 Report

The authors have revised the manuscript according to the comments and concerns of the reviewers. And then, this article could be accepted for publication of this journal, Nanomaterials.

But, the following issues are need to be checked.

 Response 9 of author: We thank the Reviewer for this important comment. Fig. 1c is the schematic diagram for testing the dielectric resistance of hybrid fibers under different conditions. In order to make reviewers more clearly understand the test route, we added the following phrases:

Line 231: As shown in Figure 1c, the hybrid fiber was placed under different conditions to test its dielectric resistance.

Line 249: In order to further explore the possible application of hybrid fibers in the damp heat sensing, as shown in Figure 1c, the hybrid fiber was placed under different conditions to test its dielectric resistance. We carried out the fluorescence tests under dry and damp heating (the result are shown in Fig. 7g-j).

 The reviewer didn’t see any statement of As shown in Figure 1c, the hybrid fiber was placed under different conditions to test its dielectric resistance in line 231 of the revised manuscript. Besides, I didn’t observe the Fig. 7g-j.
